

# Genome-wide association analysis of type II resistance to Fusarium head blight in common wheat

Dehua Wang[1,*], Yunzhe Zhao[1,*], Xinying Zhao[1], Mengqi Ji[1], Xin Guo[2], Jichun Tian[1,3], Guangfeng Chen[4] and Zhiying Deng[1]

[1] State Key Laboratory of Crop Biology, Shandong Agricultural University, Tai'an, Shandong, China
[2] Taiyuan Agro-Tech Extension and Service Center, Taiyuan, Shanxi, China
[3] Shandong Huatian Agricultural Technology Co., Ltd, Tai'an, Shandong, China
[4] College of Ecology and Garden Architecture, Dezhou University, Dezhou, Shandong, China
[*] These authors contributed equally to this work.

Corresponding author
Zhiying Deng, deng868@163.com

## ABSTRACT

**Background**. *Fusarium* head blight (FHB) is a disease affecting wheat spikes caused by some *Fusarium* species and leads to cases of severe yield reduction and seed contamination. Identifying resistance genes/QTLs from wheat germplasm may help to improve FHB resistance in wheat production.

**Methods**. Our study evaluated 205 elite winter wheat cultivars for FHB resistance. A high-density 90K SNP array was used for genotyping the panel. A genome-wide association study (GWAS) from cultivars from three different environments was performed using a mixed linear model (MLM).

**Results**. Sixty-six significant marker-trait associations (MTAs) were identified ($P < 0.001$) on fifteen chromosomes that explained the phenotypic variation ranging from 5.4 to 11.2%. Some important new MTAs in genomic regions involving FHB resistance were found on chromosomes 2A, 3B, 5B, 6A, and 7B. Six MTAs at 92 cM on chromosome 7B were found in cultivars from two different environments. Moreover, there were 11 MTAs consistently associated with diseased spikelet rate and diseased rachis rate as pleiotropic effect loci and *D_contig74317_533* on chromosome 5D was novel for FHB resistance. Eight new candidate genes of FHB resistance were predicated in wheat in this study. Three candidate genes, *TraesCS5D02G006700*, *TraesCS6A02G013600*, and *TraesCS7B02G370700* on chromosome 5DS, 6AS, and 7BL, respectively, were perhaps important in defending against FHB by regulating intramolecular transferase activity, GTP binding, or chitinase activity in wheat, but further validation in needed. In addition, a total of five favorable alleles associated with wheat FHB resistance were discovered. These results provide important genes/loci for enhancing FHB resistance in wheat breeding by marker-assisted selection.

## INTRODUCTION

Wheat (*Triticum aestivum* L.), one of the three major food crops, is grown worldwide as an essential food source and fodder. Therefore, maintaining consistent wheat production

has become a frequent focus of agricultural experts worldwide. Wheat is susceptible to both biotic (diseases, insect pests, etc.) and abiotic (drought, freezing damage, etc.) stresses because of its long growing phase. *Fusarium* head blight (FHB), also known as scab, is an infection of wheat spikes mainly caused by Fusarium species. FHB is a quantitative trait controlled by multiple genes that are affected by both the environment and genetics (*Buerstmayr & Buerstmayr, 2015*; *Liu et al., 2016b*; *Bai & Shaner, 1994*). This disease has become an important disease in the Yellow and Huai River Valleys of China (*Zhu et al., 2018*), and seriously threatens wheat production and processing. This disease not only causes severe wheat yield reduction but also contaminates wheat seeds with deoxynivalenol (DON) toxins (*Bai & Shaner, 1994*). DON, also known as vomitoxin, is regarded as a teratogen, neurotoxin and immunosuppressant, that can cause adverse health effects (*Ennouari et al., 2013*). To control human exposure to DON, the Food and Drug Administration (*Guidance for industry and FDA, 2010*) has set an advisory limit of 1 mg/kg for finished wheat products. In addition to negatively affecting wheat production in the middle and lower Yangtze River Valley region of China, FHB has become more common during the past 20 years in the Yellow and Huai River Valley regions as a result of climatic and tillage system changes. Moreover, it has become the most destructive spike disease in the world because no completely resistant varieties have been found thus far, which seriously threatens food production and food security. Breeding tolerant or fully resistant varieties and discovering resistance genes are the most effective ways to solve the problem of FHB.

The interactions between genotype and environment have a substantial impact on FHB, a complex trait with a quantitative nature. Previous research has demonstrated that FHB resistance is influenced by plant height, heading date, blooming period, extrusion, etc. The weather (sunny or wet) during blossoming is crucial for the development of this illness. Genetic linkage analysis has been used to study FHB resistance extensively in wheat, and numerous QTLs (quantitative trait loci) (more than 400 scattered on 21 chromosomes) related to FHB resistance have been reported (*Ma et al., 2020*). Five categories of FHB resistance currently exist: type I resistance to initial spike infection, type II resistance to spread spike infection, type III resistance to accumulation of mycotoxins, type IV resistance to kernel infection, and type V resistance to yield reduction (*Mesterházy, 1995*). Type I resistance and type II resistance were distinguished in the seminal study by *Schroeder & Christensen (1963)*. Almost all reports on FHB resistance have been type II. Recently, numerous small-effect Type II FHB QTLs were reviewed by *Buerstmayr, Steiner & Buerstmayr (2019)*. Seven genes (*Fhb1* to *Fhb7*) for FHB resistance have been found, and *Fhb1*, *Fhb2*, *Fhb4*, and *Fhb5* are on chromosomes 3BS, 6BS, 4BL, and 5A, respectively in common wheat; however, the remaining genes, *Fhb3*, *Fhb6,* and *Fhb7*, were derived from related wheat species (*Cainong et al., 2015*; *Jia et al., 2018*; *Li et al., 2019*; *Qi et al., 2008*; *Su et al., 2019*; *Wang et al., 2020*; *Xue et al., 2010*; *Xue et al., 2011*). The *Fhb1* gene has been widely dissected and sequenced to discover a pore-forming toxin-like (*PFT*) gene that is responsible for FHB resistance (*Rawat et al., 2016*). Later, another new gene was discovered for *Fhb1*, encoding a putative histidine-rich calcium-binding protein (*His* or *TaHRC*) that was adjacent to *PFT* (*Li et al., 2019*; *Su et al., 2019*). This research has

led to the development of *Fhb1* function markers that are being employed in molecular breeding to improve scab resistance. However, it appears that the mechanisms by which *His* and *TaHRC* impart resistance are distinct. Therefore, more research on this gene is still required to understand its molecular mechanisms (*Li et al., 2019*; *Su et al., 2019*). Recently, the candidate gene for *Fhb7* was determined and cloned; this revealed that it encoded a glutathione(GST), which can detoxify trichothecene toxins (*Wang et al., 2020*). Its resistance depends on a reduction in pathogen growth in spikes, which is different from the resistance of *Fhb1*. The remaining five FHB genes, however, have not yet been cloned.

Some significant loci for resistance have been discovered in addition to these seven FHB genes. For instance, *QFhb.mgb-2A* was identified as a *WAK2* gene (*Giancaspro et al., 2016*), and the function of *WAK2* in FHB resistance was validated (*Gadaleta et al., 2019*). Another important locus on chromosome 2DL was the transcription factor *TaWRKY70*, which regulates the expression of metabolite biosynthetic genes including *TaACT*, *TaDGK*, and *TaGLI* to influence FHB resistance (*Kage et al., 2017*; *Kage, Yogendra & Kushalappa, 2017*).Using two recombinant inbred line (RIL) populations with one common parent, named AC Barrie, from Canadian spring wheat, *QFhb.mcb-3B*, *QFhb.mcb-6B*, and *QFhb.mcb-5A.1* were mapped to the expected locations of *Fhb1*, *Fhb2*, and *Fhb5*, respectively (*Thambugala et al., 2020*). On chromosome 5B, the prominent resistance gene, *QFhb.mbr-5B* was found to explain up to 36% of the phenotypic variation (*Thambugala et al., 2020*).

With the development of genomics, growing research has been conducted on the use of genome-wide association studies (GWASs) to analyze wheat FHB resistance. One study (*Zhu et al., 2020*) used a mixed linear model(MLM) to consistently identify five quantitative trait loci (QTL) related to FHB on chromosome arms 1AS, 2DL, 5AS, 5AL, and 7DS. These QTLs accounted for 5.6, 10.3, 5.7, 5.4, and 5.6% of the variation in phenotype, respectively. *Tessmann, Dong & Van Sanford (2019)* used GWAS (based on 2-yr entry means) to identify 16 significant ($p < 0.001$) single nucleotide polymorphisms (SNPs) associated with disease traits on multiple chromosomes. Single nucleotide polymorphism association ranged from $-2.14$ to 4.01% of the mean of a given trait. Another GWAS identified 26 loci (88 marker-trait associations), which explained 6.65–14.18% of the phenotypic variances. The associated loci were distributed across all chromosomes except 2D, 6A, 6D and 7D (*Hu et al., 2020*). In addition, numerous recent studies have used GWAS in FHB resistance (*Verges, Brown-Guedira & Van Sanford, 2021*; *Gaire et al., 2021*; *Ghimire et al., 2022*; *Wu et al., 2019*). It appears that using GWAS to investigate wheat FHB is a very promising endeavor.

Therefore, our study evaluated 205 elite winter wheat accessions for FHB resistance. A high-density 90K SNP array was used to genotype the panel. A genome-wide association study (GWAS) in three environments was performed using a mixed linear model (MLM). The objective of this study was to identify some novel genomic regions associated with the type II resistance of wheat in different environments, and to predict candidate genes for loci associated with these traits, which could improve wheat FHB resistance by molecular breeding in the future.

## MATERIALS & METHODS

### Plant materials

The association mapping panel of 205 wheat accessions for GWAS comprised 77 released cultivars, 55 founder parents including two lines from Mexico and France, and 73 breeding lines from 10 provinces that represent the major winter wheat production regions in China (*Chen et al., 2016*).

Sumai 3 was selected as an FHB highly resistant check, Yangmai158 as a moderately resistant check, and Ningmaizi 22 as a moderately susceptible check.

### Growth conditions

The materials were grown in the field and greenhouse of Shandong Agricultural University (117°16′E, 36°17′N) during the 2015–2016 and 2016–2017 cropping seasons, hereafter referred to as 2016 and 2017, respectively. The terms E1, E2, and E3 represent the experimental field of Shandong Agricultural University in 2017, the greenhouse of Shandong Agricultural University in 2017, and the experimental field of Shandong Agricultural University in 2016, respectively. In the field, a randomized block design was used, with two replications. All lines were planted in 2 m plots with three rows uniformly spaced at 25 cm intervals. Each row contained 70 seeds evenly distributed seeds. The local recommended field crop management practices were followed and no pests or diseases were found in the field. However in the greenhouse, the seeds of materials were germinated and vernalized for an additional 4 weeks (4 °C, 12 h light/dark regime) before being transferred to the greenhouse. Plants were potted in a mixture of compost, sand chalk and common soil. Each plant was planted in an individual pot (the diameter was 30 cm, and the depth was 25 cm, with four seedlings) and in three replications (pots). The temperatures of the greenhouse were gradually increased from 15 °C/13 °C during day/night to 20 °C/18 °C, with a 16 h/day photoperiod at the time of anthesis. The growing conditions in the greenhouse have been described in our previous article (*Zhao et al., 2023*).

### Inoculum preparation and inoculation

In this study, the conidiospore suspension of 7136, F301, F609, and F15 virulent strains of *F. graminearum*, was obtained with the courtesy of Nanjing Agricultural University. The pathogen was propagated in mung bean medium (*Buerstmayr et al., 2000*) and incubated on a shaker at 150 rpm under 25 °C for 4–5 days. After culturing and filtering, the mass of conidia was examined under a microscope, and then, the four pathogen strains were mixed equally and stored at 4 °C for later use. The preparation of *F. graminearum* was also the same as in our previous study (*Zhao et al., 2023*).

Wheat was inoculated with 10 μl of the *F. graminearum* conidia suspension (50,000 spores/mL) applied to a pair of florets in the middle of the spikeduring flowering (*Guo et al., 2015*). Ten spikes were inoculated per line from each replicate. The whole wheat spike was then covered with a self-sealing bag to retain moisture, and had the self-sealing bag was removed after 3 days. The disease symptoms were investigated on Day 21 after inoculation, and the diseased spikelet rate (DSR), diseased spike rachis rate, and disease index (DI) were calculated. Both the field nursery and greenhouse experiments followed

this method. All spikes were classified into five classes of disease severity according to the diseased spikelet rate (DSR): 0% (class 0), 1–25% (class 1), 26–50% (class 2), 51–75% (class 3), and 76–100% (class 4) (*Lu, Cheng & Wang, 2001*). The disease index (DI) was calculated based on the rules for monitoring and forecasting wheat head blight (Chinese Standard: GB/T 15796-2011).

$$\text{Diseased spikelets rate} = \frac{\text{Number of infected spikelets}}{\text{Total spikelets per spike}} \times 100\%$$

$$\text{Diseased spike rachis rate} = \frac{\text{Number of infected rachis}}{\text{Total rachis per spike}} \times 100\%$$

$$\text{DI}(\%) = \frac{\sum h_i \times i}{H \times 4} \times 100\%$$

where $i$ is the class of disease severity, $h_i$ is the number of wheat spikes in each class, and H is the number of all investigated wheat spikes.

The standard of wheat FHB resistance was as follows: Immune (DI = 0), High resistant (DI<Sumai 3), Moderately resistant (Sumai 3<DI<Yangmai 158), Moderately susceptible (Yangmai 158<DI<Ningmaizi 22), High susceptible (DI>Ningmaizi 22).

## Genome-wide association analysis

The genome-wide positions of SNPs were used in this study based on a consensus genetic map of wheat (*Wang et al., 2014*). SNP markers, genotyping, and the population structure of the samples have been previously reported (*Chen, Chen & Tian, 2015*; *Chen et al., 2016*; *Chen et al., 2017*). A total of 24,355 mapped SNPs were used for MTA analysis. According to *Chen et al. (2017)*, the association population was divided into four categories using STRUCTURE's maximum membership probability(comprised of 43, 32, 105 and 25 varieties). *Chen et al. (2016)* also reported LD values for different chromosomes.

Based on this information, significant marker −trait associations (MTAs) were identified using a mixed linear model (MLM) in TASSEL3.0. The *P* value was used to determine whether a QTL was associated with a marker, while the phenotypic variation explained(PVE) was used to evaluate the magnitude of the MTA effects. SNPs with P $\leq 10^{-3}$ were considered to be significantly associated with phenotypic traits, and SNPs with P $\leq 10^{-4}$ were considered to be extremely significantly associated with phenotypic traits. Furthermore, when the marker was detected in two or more environments at the same time, it was considered a stable MTA.

## Statistical analysis

Analysis of variance (ANOVA) and correlations among phenotypic traits were carried out using the statistical software SPSS version 17.0 (SPSS Inc., Chicago, IL, USA).

## Forecasting candidate genes for FHB resistance

A BLAST (Basic Local Alignment Search Tool) search was performed on the International Wheat Genome Sequencing Consortium database (RefSeq v1.0; https://urgi.versailles.inra.fr/blast/) using the sequence of the significant SNP markers identified by GWAS. When

**Table 1  Phenotypic variation of wheat diseased spikelets rate.**

| Environment | Change | Mean | Standard deviation | Coefficient of variation |
|---|---|---|---|---|
| E1 | 0.0611~1 | 0.7721 | 0.28 | 36.55% |
| E2 | 0.0477~1 | 0.6388 | 0.34 | 52.96% |
| E3 | 0.0111~1 | 0.7494 | 0.25 | 44.30% |

**Notes.**

E1, the experimental field of Shandong Agricultural University in 2017; E2, the greenhouse of Shandong Agricultural University in 2017; E3, the experimental field of Shandong Agricultural University in 2016.

an SNP marker sequence from the IWGSC was 100% identical to any wheat contig, the sequence was extended by 2 Mb for each marker using the IWGSC BLAST results. Afterward, the extended sequence was used to run a BLAST search on the National Center for Biotechnology Information (NCBI) database (https://www.ncbi.nlm.nih.gov/) and on Ensembl Plants (http://plants.ensembl.org/Triticum_aestivum/Tools/Blast) to confirm possible candidate genes and functions.

## Analysis of marker haplotype

The effect of resistance genes was calculated by using the average diseased spikelet rate of various gene combinations. Effect of resistance genes = (Average diseased spikelet rate of materials carrying resistance gene − Average diseased spikelet rate of materials without resistance gene)/Average diseased spikelet rate of materials without resistance gene.

## RESULTS

### Phenotypic variation analysis of wheat FHB resistance

The variation coefficient of the diseased wheat spikelet rate (DSR) was the highest in E2 (52.96%), followed by that in E3 (44.30%), and E1(36.55%) (Table 1), indicating that DSR genetic variation was abundant. The variance analysis of FHB resistance of the spikelet and spike rachis indicated that significant differences were present between cultivars and environments, and their interactions (Table 2). This illustrated that FHB resistance was a quantitative trait affected not only by genotype but also by the environment. Furthermore, there were significant positive correlation coefficients between the spikelet and spike rachis in the three environments, indicating that the resistance trend of FHB was consistent between the spikelet and spike rachis (Table 3).

### Marker–trait associations (MTAs) of FHB resistance

Sixty-six MTAs associated with FHB resistance were distributed on chromosomes 1A, 1B, 2A, 2B, 2D, 3B, 3D, 4A, 5A, 5B, 5D, 6A, 6B, 7A, and 7B (Tables S1 and S2; Fig. 1). The phenotypic variation explained(PVE) of MTA loci to FHB resistance ranged from 5.45% to 11.20%, of which, 11 MTA loci were detected in both the spikelet and spike rachis. On chromosome 7B, a novel genomic region from genetic position 92 to 103, significantly associated with FHB resistance, was detected in all three environments. Moreover, there was one major locus at genetic position 92 of chromosome 7B accounting for 11.20% of the phenotypic variation in the spikelets, namely locus *BS00025286_51*, which was also

**Table 2  ANOVA of wheat diseased spikelet and spike rachis rate in different environments.**

| Source | Dependent variable | Type III Sum of Squares | Degree of freedom | Mean Square | *F*- value |
|---|---|---|---|---|---|
| Varieties | Spikelet | 94.062 | 204 | 0.461 | 16.421[*] |
| | Spike rachis | 88.842 | 204 | 0.435 | 19.698[*] |
| Environment | Spikelet | 5.262 | 2 | 2.631 | 93.709[*] |
| | Spike rachis | 5.505 | 2 | 2.752 | 124.493[*] |
| Varieties * Environment | Spikelet | 85.307 | 406 | 0.21 | 7.483[*] |
| | Spike rachis | 83.735 | 406 | 0.206 | 9.329[*] |
| Error | Spikelet | 34.425 | 1226 | 0.028 | |
| | Spike rachis | 27.105 | 1226 | 0.022 | |
| Total | Spikelet | 1128.175 | 1839 | | |
| | Spike rachis | 1182.519 | 1839 | | |

**Notes.**

[*]Significant at the 0.05 level (2-tailed).

**Table 3  The correlation coefficients of spikelet and spike rachis in three environments, respectively.**

| Environment[a] | Variable | E1 | | E2 | | E3 | |
|---|---|---|---|---|---|---|---|
| | | Spikelet | Spike rachis | Spikelet | Spike rachis | Spikelet | Spike rachis |
| E1 | Spikelet | 1 | | | | | |
| | Spike rachis | 0.881[**] | 1 | | | | |
| E2 | Spikelet | 0.318[**] | 0.203[**] | 1 | | | |
| | Spike rachis | 0.355[**] | 0.239[**] | 0.902[**] | 1 | | |
| E3 | Spikelet | 0.263[**] | 0.205[**] | 0.224[**] | 0.142[**] | 1 | |
| | Spike rachis | 0.233[**] | 0.202[*] | 0.205[**] | 0.118[*] | 0.986[**] | 1 |

**Notes.**

[a]E1, E2 and E3 were the same as the Table 1.

[**]Correlation is significant at the 0.001 level(2-tailed)

[*]Correlation is significant at the 0.05 level(2-tailed).

detected for the spike rachis, explaining 7.07% of its phenotypic variation. In E3, four loci on chromosome 7B were found to be associated with both diseased spikelet rate and diseased spike rachis rate, but these MTAs are located in the same region and may represent one QTL (Table 4; Table S2). In addition, there were some genomic regions associated with FHB resistance on chromosomes 5B, 6A, 2A, and 3B, but they were found only in a single environment. The other six loci, including *D_contig74317_533* on chromosome 5D, *Kukri_c14239_1995* on chromosome 6A, *Kukri_c7087_896* on chromosome 3B, *RAC875_c35801_905* on chromosome 3D, *BS00099729_51* on chromosome 5B, and *RAC875_c68525_284* on chromosome 6B, were also determined to be associated with both the diseased spikelet rate and the diseased spike rachis rate. The remaining MTA loci were detected only for a single trait in a single environment.

## Allelic variation analysis of MTA loci

According to the PVE of MTA loci (Table S2), we selected 10 of markers and analyzed their allelic variation (Table 4). Alleles T and C of the marker, *Kukri_c14239_1995* on chromosome 6A were associated with the largest phenotypic difference (0.2297). Specifically,

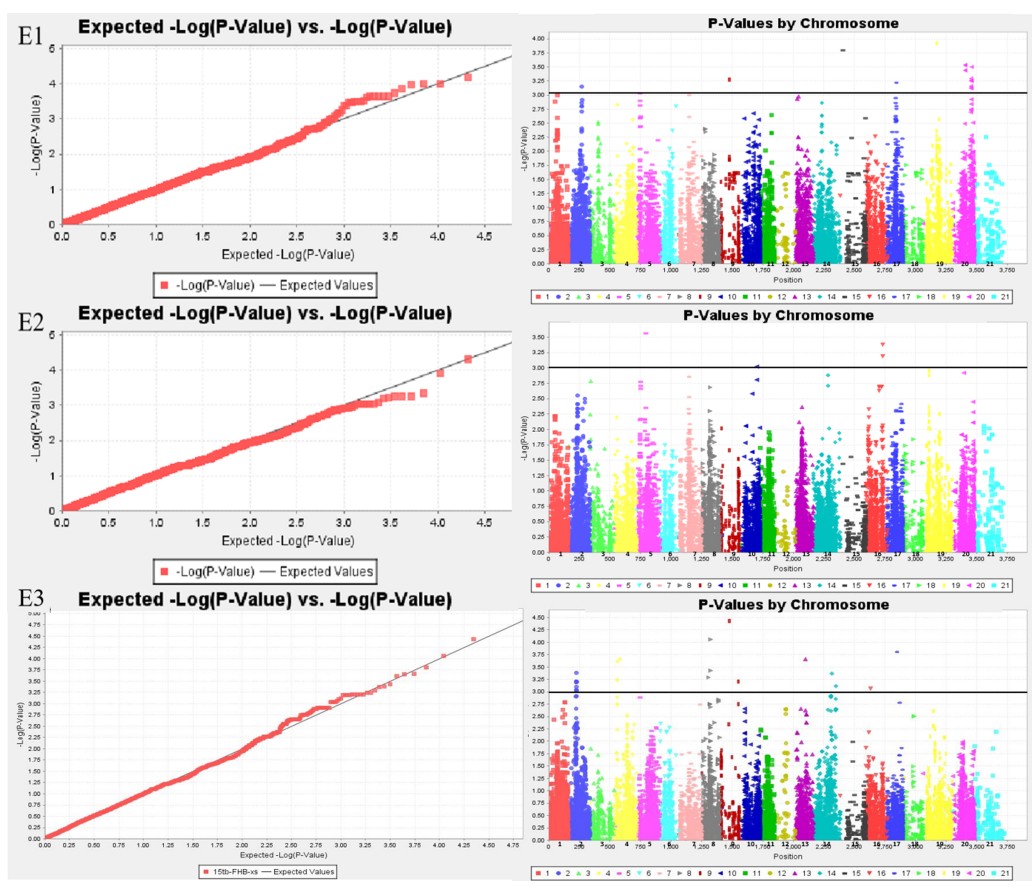

**Figure 1** **Whole genome association analysis QQ map (left) and Manhattan plot of disease spikelet rate (right).** E1: the experimental field of Shandong Agricultural University in 2017; E2: the greenhouse of Shandong Agricultural University in 2017; E3: the experimental field of Shandong Agricultural University in 2016. 1-21: 1A, 1B, 1D, 2A, 2B, 2D, 3A, 3B, 3D, 4A, 4B, 4D, 5A, 5B, 5D, 6A, 6B, 6D, 7A, 7B, 7D.

the phenotypic value of the diseased spikelet rate associated with *Kukri_c14239_1995-T* was significantly higher than that associated with *Kukri_c14239_1995-C*, indicating that *Kukri_c14239_1995-C* was better than *Kukri_c14239_1995-T* for FHB resistance (Table 4). Furthermore, because allele C of *D_contig74317_533* showed a significantly higher diseased spikelet rate than *D_contig74317_533-T*, allele T was deemed to be better for improving FHB resistance. On chromosome 7B, allele C of *BS00025286_51* had a higher diseased spikelet rate than allele T; thus, allele T for this locus was favorable for FHB resistance. Nevertheless, for the other four loci on this chromosome, significant differences between the two alleles for the diseased spikelet rate seemed to be at 5%. The smallest difference in diseased spikelet resistance was observed between *Kukri_c7087_896-G* and *Kukri_c7087_896-A*, which indicated that this locus affected FHB resistance to a smaller degree. Moreover, on chromosome 3D, *RAC875_c35801_905-G* yielded better results than *RAC875_c35801_905-A* for FHB resistance.

**Table 4  Phenotypic effect of alleles for the relatively stable loci of disease spikelet rate.**

| Locus | Chromosome | Allele | Variety number | Environment[a] | | | Average | Difference[b] |
|---|---|---|---|---|---|---|---|---|
| | | | | E1 | E2 | E3 | | |
| D_contig74317_533 | 5D | CC | 177 | 0.7829 | 0.7585 | 0.764 | 0.7685Aa | 0.0989 |
| | | TT | 28 | 0.6983 | 0.6519 | 0.6586 | 0.6696Bb | |
| Kukri_c14239_1995 | 6A | TT | 192 | 0.795 | 0.754 | 0.762 | 0.7703Aa | 0.2297 |
| | | CC | 11 | 0.4593 | 0.619 | 0.5435 | 0.5406Bb | |
| BS00025286_51 | 7B | CC | 125 | 0.8001 | 0.7584 | 0.7484 | 0.7689a | 0.096 |
| | | TT | 52 | 0.6529 | 0.6632 | 0.7025 | 0.6729b | |
| Kukri_c7087_896 | 3B | GG | 14 | 0.8014 | 0.8043 | 0.7106 | 0.7721a | 0.0185 |
| | | AA | 191 | 0.7691 | 0.7393 | 0.7524 | 0.7536a | |
| RAC875_c35801_905 | 3D | AA | 186 | 0.7841 | 0.7521 | 0.7547 | 0.7636Aa | 0.094 |
| | | GG | 19 | 0.6463 | 0.6635 | 0.699 | 0.6696Bb | |
| RAC875_c68525_284 | 6B | AA | 22 | 0.7774 | 0.7489 | 0.7525 | 0.7596a | 0.0437 |
| | | GG | 183 | 0.7209 | 0.7018 | 0.725 | 0.7159b | |
| Kukri_c4143_1055 | 7B | AA | 104 | 0.7907 | 0.7624 | 0.7661 | 0.7731a | 0.0379 |
| | | CC | 101 | 0.7505 | 0.7244 | 0.7306 | 0.7352b | |
| RAC875_c18043_369 | 7B | CC | 104 | 0.7915 | 0.763 | 0.768 | 0.7741a | 0.0402 |
| | | TT | 100 | 0.7502 | 0.7229 | 0.7287 | 0.7339b | |
| RAC875_c18043_411 | 7B | AA | 105 | 0.7914 | 0.764 | 0.7695 | 0.7749a | 0.041 |
| | | GG | 100 | 0.7502 | 0.723 | 0.7287 | 0.7339b | |
| RAC875_c5646_774 | 7B | GG | 104 | 0.7915 | 0.763 | 0.768 | 0.7742a | 0.0391 |
| | | AA | 101 | 0.7505 | 0.7244 | 0.7306 | 0.7351b | |

**Notes.**
[a] E1, E2 and E3 were same as Table 1.
[b] Difference between alleles. A and B: Different capital letters indicate significant difference between alleles at one locus at $P \leq 0.01$; a & b: Different lowercase letters indicate significant difference between alleles at one locus at $P \leq 0.05$

## Prediction of candidate genes for some important loci

Eight important candidate genes were screened for important loci significantly associated with the diseased spikelet rate and diseased spike rachis rate in wheat (Table S3). One candidate gene, *TraesCS3D02G326700* located on chromosome 3D is associated with actin-binding in wheat. The candidate gene, *TraesCS5D02G006700* of the marker, *D_contig74317_533* on 5DS was predicted in wheat, whose function was intramolecular transferase activity. Two candidate genes *TraesCS6A02G013700* and *TraesCS6A02G013800*, predicted by *IAAV9150*, participate in the ubiquitin-dependent ER-associated degradation (ERAD) pathway in wheat. The candidate gene, *TraesCS6A02G013600* of *Excalibur_c20597_509* functions in GTP binding in wheat. On chromosome 7BL, *TraesCS7B02G370700* of *BS00025286_51* is involved in the biological process of defense response to fungi. There is one candidate gene, *TraesCS7B02G340200*, for the three loci, *RAC875_c18043_369*, *RAC875_c18043_411*, and *Kukri_c4143_1055*, on chromosome 7BL that was identified because of being in the same physical location. The candidate gene, *TraesCS7B02G340100* of *RAC875_c5646_774* is associated with the carbohydrate metabolic process in *Triticum aestivum*. By analyzing the homologous genes of these candidate genes, we found that the functions or biological processes of most

homologous genes in other crops, such as *Japonica rice*, *Hordeum vulgare*, *Oryza sativa Indica*, *Oryza sativa Japonica*, and *Arabidopsis thalian*, are involved in the defense response of fungi, which indicates that these candidate genes may be related to FHB resistance, but this needs further verification in wheat.

### Analysis of marker haplotype and resistance

Among the alleles at the marker −trait associations, the alleles with decreasing diseased spikelet rate were assumed to be the resistance alleles at this site. In this study, five important SNP loci were selected to evaluate the effect of aggregation of their favorable alleles on decreasing the diseased spikelet rate (Table 5). In general, with the increase in the number of favorable alleles, the effect of reducing the rate of diseased spikelets was more obvious, which could improve FHB resistance. By gene combination analysis (Table 6), two samples, B202 and B34, had four favorable alleles with TCTAC and TTTGC haplotypes, respectively. The haplotype TCTAC showed high resistance to FHB (Table 6). Six haplotypes with three favorable alleles were found. The materials with the haplotypes TCCGA and TTTAC showed high resistance to FHB, including B70, B72 and B179. The residue haplotypes showed moderate resistance or moderate susceptibility to FHB. This result indicated that multiple haplotypes of the materials played an important role in the screening of anti-scab materials. In addition, by comparing the reported GWAS results for plant height using the same panel materials (*Chen, Chen & Tian, 2015*), these five loci had no significant effect on plant height (Tables S4 and S5).

## DISCUSSION

The majority of earlier researchers focused on finding the significant genes/loci of FHB *via* QTL mapping or association mapping to improve the resistance to FHB in wheat cultivars (*Löffler, Schön & Miedaner, 2009*; *Li et al., 2011*; *Venske et al., 2019*). Previous findings suggested that practically all wheat chromosomes were implicated (*Yu, 2007*; *Ma et al., 2020*), but chromosomes 3B, 4B, 5A, and 6B seemed to be important because of the *Fhb1*, *Fhb2*, *Fhb4*, and *Fhb5* genes (*Löffler, Schön & Miedaner, 2009*; *Zhang et al., 2018*). Five QTLs were discovered by GWAS analysis on chromosome arms 1AS, 2DL, 5AS, 5AL, and 7DS (*Zhu et al., 2020*). Of these, the locus on chromosome 5B, for decreasing the amount of deoxynivalenol may be novel. However, in our study, fifteen chromosomes were involved in the MTA loci, and some important genomic regions involved in FHB resistance were found on chromosomes 2A, 3B, 5B, 6A, and 7B. The significance of chromosome 3B for FHB resistance was further supported by this result. Six MTAs at 92 cM were also discovered on chromosome 7B in two different contexts. Of these, the BS00025286 51 locus explained 11.20% of the phenotypic variation and five MTAs were consistently associated with the diseased spikelet rate and diseased rachis rate. It appeared that FHB resistance was relevant in this region. More intriguingly, five SNP markers(*Kukri_c14239_1995* on chromosome 6A, *Kukri_c7087_896* on chromosome 3B, *RAC875_c35801_905* on chromosome 3D, *BS00099729_51* on chromosome 5B, and *RAC875_c68525_284* on chromosome 6B) in this study have appeared in previous reports (*Venske et al., 2019*), which indicated the reliability

Wang et al. (2023), *PeerJ*, DOI 10.7717/peerj.15906

**Table 5  Disease resistance statistics of different resistant QTL combinations.**

| | D_contig74317_533 | Kukri_c14239_1995 | BS00025286_51 | RAC875_c35801_905 | Kukri_c4143_1055 | Variety number | Mean DSR | Effect of resistance genes | Standard deviation | F-value | P-value |
|---|---|---|---|---|---|---|---|---|---|---|---|
| | − | − | − | − | − | 38 | 0.848 | . | 0.148 | | |
| | + | − | − | − | − | 9 | 0.866 | 0.022 | 0.081 | | |
| | − | + | − | − | − | 3 | 0.815 | −0.038 | 0.088 | | |
| | − | − | + | − | − | 28 | 0.713 | −0.159 | 0.237 | | |
| | − | − | − | + | − | 2 | 0.922 | 0.087 | 0.047 | | |
| | − | − | − | − | + | 62 | 0.777 | −0.084 | 0.230 | | |
| | + | − | + | − | − | 1 | 0.725 | −0.145 | . | | |
| | + | − | − | − | + | 1 | 0.330 | −0.611 | . | | |
| | − | − | + | + | − | 2 | 0.905 | 0.068 | 0.089 | | |
| | − | − | + | − | + | 10 | 0.684 | −0.193 | 0.343 | | |
| Genotype | − | − | − | + | + | 2 | 0.892 | 0.052 | 0.051 | 2.551 | 0.001 |
| | + | + | − | + | − | 2 | 0.186 | −0.781 | 0.017 | | |
| | + | + | − | − | + | 1 | 0.726 | −0.144 | . | | |
| | + | − | + | − | + | 2 | 0.498 | −0.413 | 0.491 | | |
| | + | − | − | + | + | 3 | 0.609 | −0.281 | 0.304 | | |
| | − | + | + | + | − | 2 | 0.617 | −0.272 | 0.068 | | |
| | − | + | + | − | + | 1 | 0.844 | −0.005 | . | | |
| | − | − | + | + | + | 3 | 0.593 | −0.300 | 0.270 | | |
| | + | + | + | − | + | 1 | 0.210 | −0.752 | . | | |
| | + | − | + | + | + | 1 | 0.40 | −0.524 | . | | |

**Notes.**

"+" represents the allele for improving scab resistance; "–" represents the allele that reduces the resistance.

**Table 6  Haplotype of marker associated with FHB resistance and corresponding carrier materials.**

| D_contig74317_533 | Kukri_c14239_1995 | BS00025286_51 | RAC875_c35801_905 | Kukri_c4143_1055 | Number of resistance alleles | Variety | Disease spikelet rate | FHB resistance |
|---|---|---|---|---|---|---|---|---|
| T | C | C | G | A | 3 | B70 | 0.1979 | HR |
|   |   |   |   |   |   | B72 | 0.1739 | HR |
| T | C | C | A | C | 3 | B97 | 0.7256 | MS |
| T | T | T | A | C | 3 | B179 | 0.1501 | HR |
| T | T | C | G | C | 3 | B131 | 0.3217 | MR |
|   |   |   |   |   |   | B200 | 0.5782 | MS |
| C | C | T | G | A | 3 | B44 | 0.6654 | MS |
|   |   |   |   |   |   | B196 | 0.5691 | MS |
| C | T | T | G | C | 3 | B16 | 0.3519 | MR |
|   |   |   |   |   |   | B68 | 0.5425 | MS |
| T | C | T | A | C | 4 | B202 | 0.2100 | HR |
| T | T | T | G | C | 4 | B34 | 0.4037 | MS |

of our results. There was one novel locus *D_contig74317_533* on chromosome 5D found for FHB resistance.

According to previous research, plant height had an impact on FHB resistance under field conditions. QTL mapping showed that approximately 40% of the QTLs for plant height overlapped with the QTLs for FHB resistance on 14 chromosomes (*Buerstmayr, Steiner & Buerstmayr, 2019*). Five QTLs for FHB resistance were discovered by *Zhu et al. (2020)*, among which *QFhb.hbaas-5AS* had a significant correlation with plant height. Thirty-eight MTA loci for plant height were discovered on chromosomes 1B, 2A, 2B, 3A, 3B, 3D, 4A, 4B, 5A, and 6D by *Chen, Chen & Tian (2015)* using a panel of 205 wheat accessions for the GWAS analysis of plant height. Of these, 11 loci were detected on chromosome 6D in two or more environments (*Chen, Chen & Tian, 2015*). However in this study, there were five MTA loci for FHB resistance that showed no significant relationship with plant height, that is, *D_contig74317_533* on chromosome 5D, *Kukri_c14239_1995* on 6A, *RAC875_c35801_905* on 3D, and *Kukri_c4143_1055* and *BS00025286_51* on 7B, which indicated that these loci can be flexibly used in breeding. Additionally, wheat materials with the aforementioned loci were screened, and by using molecular marker-assisted selection, they can be incorporated into major Chinese planted varieties without compromising plant height.

In fact, plant disease resistance is a complex molecular process controlled by genes (*Ma et al., 2020*). Only a few significant genes were found, although researchers have discovered hundreds of QTLs scattered across wheat, including 21 chromosomes from common wheat varieties or related species (*Buerstmayr, Ban & Anderson, 2009*; *Liu et al., 2009*). Nevertheless, it has become simpler to find additional genes as a result of the advances in molecular technology and the wheat genome sequence (both major and minor). Moreover, the isolation and functional verification of FHB resistance genes are beneficial to understanding the pathogenesis and resistance mechanism of wheat FHB at the molecular level (*Liu et al., 2016b*). Previous research has demonstrated that the mechanism of the genes/loci identified in FHB resistance could involve a complicated signal transduction pathway and be associated with the synergistic effect of many protein factors (*Zhang et al., 2018*; *Liu et al., 2016a*; *Dweba et al., 2017*). For example, the genes encoding a 12-oxophytodienoate reductase-like protein identified in the region of *QFh.hbaas-1AS* may be related to the biosynthesis or metabolism of signaling molecules, oxylipins, such as jasmonic acid (JA) (*Ding et al., 2011*; *Qi et al., 2016*). These genes were discovered to encode several different proteins, including receptor-like kinase, UDP-glycosyltransferase, pathogenesis-related protein 1, and glucan endo-1,3-beta-glucosidase (PR2) (*Anand et al., 2003*; *Pan et al., 2018*; *Ma et al., 2020*).

In this study, the candidate genes on chromosome 3D encoded UDP-glycosyltransferase activity and were related to the defense response to biotic stimulus (*Li et al., 2017*). This indicated that this gene may enhance resistance to FHB because this protein could detoxify both DON and NIV produced by *F. graminearum* (*Poppenberger et al., 2003*; *Zhu et al., 2020*). By performing homologous gene detection on the *D_contig74317_533* locus of chromosome 5D, the genes were found to have homologous nucleic acid binding and defense functions in barley, *Arabidopsis*, indica rice, japonica rice, and wild rice. Of these,

the *AT2G39510* gene is related to the activity of glutamine transmembrane transporter protein. Previous studies have shown that the glutamine-gated ion channel is related to the function of the *Fhb5* gene, which can control $Ca^{2+}$ influx (*Dennison, 2000*; *Kugler et al., 2013*). It was also found that $Ca^{2+}$ was involved in early signaling defense against FHB (*Ding et al., 2011*). Recent research has revealed that wall-associated kinase(WAK) is a receptor-like protein kinase, that is involved in signal transduction and the defense response of plants (*Zhang et al., 2006*; *Hu & Lo, 2010*). In this study, it was shown that the gene *Traescs6A02G013600* contains homologous genes in Arabidopsis and Japonica rice, some of which encode members of the receptor-like cytoplasmic kinase (RLCK) and wall-associated kinase (WAK) families. Therefore, it is possible that this gene contributes to wheat FHB resistance, but more research is needed. Additionally, earlier research has demonstrated that the pathogenesis-related protein (PR) chitinase participates in the plant's fundamental defense mechanism and starts to accumulate during pathogen infection (*Ma et al., 2020*). Fortunately, the gene, *TraesCS7B02G370700* of *BS00025286_51* on the 7BL chromosome was also found to be associated with chitinase activity and the defense response for fungi in our study. Meanwhile, the eight candidate genes identified were associated with either calcium ion binding or GTP binding, which has been shown to be involved in the early response of wheat to *F. graminearum* infection by salicylic acid (SA) and $Ca^{2+}$ signals (*Ding et al., 2011*). Given that $Ca^{2+}$ signal transduction was discovered to be crucial for the transcriptional reprogramming of innate plant immunity (*Boudsocq et al., 2010*), it is possible that these candidate genes associated with $Ca^{2+}$ signals will be crucial in protecting against FHB in wheat.

## CONCLUSIONS

In this study, sixty-six significant marker-trait associations (MTAs) were identified ($P<0.001$) on fifteen chromosomes, explaining phenotypic variation ranging from 5.4% to 11.2%. Some genomic regions involved in FHB resistance were found on chromosomes 2A, 3B, 5B, 6A and 7B. There were eleven MTAs consistently associated with the disease spikelet rate and disease rachis rate as pleiotropic effect loci. Eight new candidate genes of FHB resistance were predicted in wheat. Of these, three genes *TraesCS5D02G006700*, *TraesCS6A02G013600 and TraesCS7B02G370700* on chromosomes 5DS, 6AS and 7BL, respectively, possibly defending against FHB by regulating intramolecular transferase activity, GTP binding, and chitinase activity in wheat. In addition, a total of five favorable alleles associated with wheat FHB resistance were discovered in this study. In the materials with multiple favorable alleles, the resistance was mostly moderately resistant or moderately susceptible.

## ACKNOWLEDGEMENTS

We thanked Dr. Jirui Wang from Sichuan Agricultural University and Dr. Haiyan Jia from Nanjing Agricultural University for analyzing SNP genotyping and providing virulent strains of *F. graminearum*, respectively.

### Funding

This work was supported by the Agricultural Improved Variety Project of Shandong Province (No. 2022LZGCQY002), the Agricultural Improved Variety Project of Tai'an City (2022NYLZ06), and the Shandong Province Postgraduate Education Tutor Capacity Improvement Program (No. SDYY18114). The funders had no role in study design, data collection and analysis, decision to publish, or preparation of the manuscript.

### Grant Disclosures

The following grant information was disclosed by the authors:
Agricultural Improved Variety Project of Shandong Province: 2022LZGCQY002.
Agricultural Improved Variety Project of Tai'an City: 2022NYLZ06.
Shandong Province Postgraduate Education Tutor Capacity Improvement Program: SDYY18114.

### Competing Interests

Jichun Tian is employed by Shandong Huatian Agricultural Technology Co., Ltd. The authors have no competing interests.

### Author Contributions

- Dehua Wang performed the experiments, analyzed the data, prepared figures and/or tables, and approved the final draft.
- Yunzhe Zhao performed the experiments, analyzed the data, prepared figures and/or tables, and approved the final draft.
- Xinying Zhao performed the experiments, analyzed the data, prepared figures and/or tables, and approved the final draft.
- Mengqi Ji performed the experiments, analyzed the data, prepared figures and/or tables, and approved the final draft.
- Xin Guo performed the experiments, analyzed the data, prepared figures and/or tables, and approved the final draft.
- Jichun Tian analyzed the data, authored or reviewed drafts of the article, and approved the final draft.
- Guangfeng Chen performed the experiments, analyzed the data, prepared figures and/or tables, and approved the final draft.
- Zhiying Deng conceived and designed the experiments, performed the experiments, authored or reviewed drafts of the article, and approved the final draft.

### Data Availability

The raw data is available in the Supplemental Files.

### Supplemental Information

Supplemental information for this article can be found online at http://dx.doi.org/10.7717/peerj.15906#supplemental-information.

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
