# Peer review of "Genome-wide association analysis of type II resistance to Fusarium head blight in common wheat"

_PeerJ, doi:10.7717/peerj.15906_

## Round 0.1 · original submission · Major Revisions

Dear Author

The reviewers have recommended revisions to your manuscript. Therefore, I invite you to respond to the reviewer's comments and revise your manuscript.

With Thanks

·

Basic reporting

This paper is a nice piece of work, notably presenting a major QTL for resistance to FHB detected on the chromosome 7B. However I suggest some minor changes:
Introduction: A bit more details on FHB fungi would be good. And are there studies detecting QTLs using the same F. graminearum strains than in your study?
Table S1: Which genetic map is used?
Table S2: Precise the version of the physical map (RefSeq v2.1?)
Table S4: Could you add p-values?
Figure 1 and Table S1 are not matching properly -> indeed the QTL on 7B is detected in all the environments in Table S1, but not on Figure 1 …
Raw data1 and 3: sheet names are in Chinese as well as variety names > is it possible to translate?
I wrote many other small comments in the text.

Experimental design

The experimental set up and phenotyping are well described. However, for the GWAS analysis, MTAs are detected on 15 chromosomes, but it is unclear how they were prioritised.
It is nice to have investigated potential effects of plant height.

Validity of the findings

The GWAS results seems reliable and are very interesting. More discussion of the detection of already known QTL could support the validity of the findings: are the QTL expected to be present in the Chinese germplasm segregated in this collection?
However, the method for the selection of the very short list of candidate genes is unclear.

Additional comments

It is very interesting to have the new QTL on chromosome 7B, and reliably and strongly detected in all the environments, most importantly for the disease spikelet rate. It would be good to emphasize the abstract a bit more on this QTL, for the abstract to be more related to the title. Is this QTL linked to one of the subgroups of your collection?

Reviewer 2 ·

Basic reporting

Significant improvement in writing is necessary

Experimental design

Need more explanations

Validity of the findings

Comments included in the pdf file

Annotated reviews are not available for download in order to protect the identity of reviewers who chose to remain anonymous.

Reviewer 3 ·

Basic reporting

The manuscript “Identification of a novel genomic region associated with resistance to Fusarium head blight in Chinese winter wheat” reports the identification of markers associated to resistance to Fusarium head blight in common wheat. The writing is clear, and I do not see any concern about literature references and the structure of the article.

Experimental design

A panel of 205 genotypes has been used, with a phenotypic evaluation carried out in three environments (two in field and one in controlled conditions) with two biological replications. The experimental design seems adequate for the study.

Validity of the findings

The study is of interest, but there some concerns to be addressed.
First of all, all the MTAs should be grouped in QTLs based on effect and average extension of linkage disequilibrium. Once identified the single QTLs, these should be reported on a physical map of wheat chromosomes for a clear visualization.
The results about single MTAs are not so convincing, especially based on QQ-plots, which show the absence of P values above the diagonal of expected values. Indeed, the threshold to declare significant MTAs is chosen arbitrarily by the authors without any correction usually used in GWAS as Bonferroni or FDR. Furthermore, the additive effect of single MTAs is very low, indicating that single QTLs are not so useful in improving resistance. A clear effect is observed only when lines containing the positive allele at many QTLs at the same time are considered.
The author should try some other GWAS models (MLM+structure or kinship, or other models integrated by GAPIT) and show the corresponding QQ-plots, maybe different models can lead to identify stronger associations.

---

## Round 0.2 · Minor Revisions

Dear Authors

According to the reviewer's comments, the manuscript needs a minor revision to be reconsidered for publication. The authors are invited to revise the paper considering all the suggestions made by the reviewers. Please note that requested changes are required for publication.

Best Regards

·

Basic reporting

A GWAS analysis for FHB resistance is very useful to decipher the genetic basis of this trait. The context and experiments are well presented and the manuscript have been improved since the last submission.
I wrote a few comments in the manuscript.

Comments on the supplementary data:
Rawdata1.xls: It would be nice to add the status cultivars/breeding lines and the origin of the lines (which ones are from Mexico and France?)
Rawdata2.xls: This file doesn’t contain any text on how to read it + the code of the lines are different than in the previous file.
Table S1: Add reference to the genetic map: Wang et al. 2014
Table S4: Could you add p-values?

Experimental design

The experimental design is well described.

Validity of the findings

The results presented in the table S1 are different than on the Figure 1, particularly for E3, where the most important peak on Figure 1 is on chromosome 6A, which do not appear on table S1 for the same trait. Similarly, the SNPs detected on chromosome 7B on table S1 also for E3 are not appearing on Figure 1. Which results are the correct results? I guess it is just an issue with Figure 1?
It is important that the results presented on Figure 1 and Table S1 for the same trait are identical ...

---

## Round 0.3 · Minor Revisions

Dear Authors
Significant concerns about the manuscript's grammar, usage, and overall readability exist. We, therefore, request that you revise the text to fix the grammatical errors and improve the overall readability of the text.


Here is an example of how I would modify the abstract:

Background. Fusarium head blight (FHB) is a disease affecting wheat spikes caused by some Fusarium species and leads to cases of severe yield reduction and seed contamination. Identifying resistance genes/QTLs from wheat germplasm may help to improve FHB resistance in wheat production. Methods. Our study evaluated 205 elite winter wheat cultivars for FHB resistance. A high-density 90K SNP array was used for genotyping the panel. A genome-wide association study (GWAS) from cultivars from three different environments was performed using a mixed linear model (MLM). Results. Sixty-six significant marker–trait associations (MTAs) were identified (P<0.001) on fifteen chromosomes that explained the phenotypic variation ranging from 5.4 to 11.2%. Some important new MTAs in genomic regions involving FHB resistance were found on chromosomes 2A, 3B, 5B, 6A, and 7B. Six MTAs at 92 cM on chromosome 7B were found in cultivars from two different environments .
Moreover, there were 11 MTAs consistently associated with diseased spikelet rate and diseased rachis rate as pleiotropic effect loci and D_contig74317_533 on chromosome 5D was novel for FHB resistance. Eight new candidate genes of FHB resistance were predicated in wheat in this study. Three candidate genes, TraesCS5D02G006700, TraesCS6A02G013600, and TraesCS7B02G370700 on chromosome 5DS, 6AS, and 7BL, respectively, were perhaps important in defending against FHB by regulating intramolecular transferase activity, GTP binding, or chitinase activity in wheat, but further validation is needed. In addition, a total of five favorable alleles associated with wheat scab resistance were discovered. These results provide important genes/loci for enhancing FHB resistance in wheat breeding by marker-assisted selection.

The remainder of the manuscripts should be edited accordingly."

With Thanks

---

## Round 0.4 · Minor Revisions

Dear Author
According to the reviewer's comments, this manuscript needs a minor revision to be reconsidered for publication. The authors are invited to revise the paper considering all the suggestions made by the reviewer. Please note that requested changes are required for publication.
Regards

·

Basic reporting

The article was very much improved since the last review and is now clearer and more complete.

Experimental design

The experimental design is well described.

Validity of the findings

I still have an issue with the results presented on Figure 1 different than those presented on the table S1. The figure 1 has now been corrected for the environment E3, but in the case of E1 the most significant MTA appears to be on chromosome 5A on figure 1, while on table S1, this chromosome do not have any significantly associated SNP ...
In addition, in your conclusion, for the sentence starting with "Of these, three genes TraesCS5D02G006700, TraesCS6A02G013600381 and TraesCS7B02G370700", I suggest you could be a bit more moderate regarding the involvement of these genes, as they are closeby the associated markers, but other genes were also in the 2Mb intervals around the associated markers you looked at, and the genome annotation/assembly does not necessarily contain the causative gene for the resistance.

Additional comments

Overall, the article is well written and will be valuable to the study of the genetic basis of wheat resistance to FHB.

---

## Round 0.5 · accepted · Accept

Dear Authors,

I am pleased to inform you that after the last round of revision, the manuscript has been improved a lot, and it can be accepted for publication.

Congratulations on accepting your manuscript, and thank you for your interest in submitting your work to PeerJ.

With Thanks

·

Basic reporting

Article clear and well written.

Experimental design

A lot of experimental work has been necessary and is well described.

Validity of the findings

This GWAS will be a valuable contribution to the study of FHB resistance in wheat.